# Cell invasion during competitive growth of polycrystalline solidification patterns

Younggil Song [1,2], Fatima L. Mota [3], Damien Tourret[4], Kaihua Ji[1], Bernard Billia[3], Rohit Trivedi[5], Nathalie Bergeon[3] & Alain Karma [1] ✉

Spatially extended cellular and dendritic array structures forming during solidification processes such as casting, welding, or additive manufacturing are generally polycrystalline. Both the array structure within each grain and the larger scale grain structure determine the performance of many structural alloys. How those two structures coevolve during solidification remains poorly understood. By in situ observations of microgravity alloy solidification experiments onboard the International Space Station, we have discovered that individual cells from one grain can unexpectedly invade a nearby grain of different misorientation, either as a solitary cell or as rows of cells. This invasion process causes grains to interpenetrate each other and hence grain boundaries to adopt highly convoluted shapes. Those observations are reproduced by phase-field simulations further demonstrating that invasion occurs for a wide range of misorientations. Those results fundamentally change the traditional conceptualization of grains as distinct regions embedded in three-dimensional space.

Boundaries emerge in a wide range of pattern-forming systems[1], e.g. during Rayleigh–Bénard convection[2], biological tissue development[3–5], tumor growth with cells migrating in foreign tissues[6,7], self-assembly during the growth of colloidal crystals[8,9], polycrystalline growth in polymeric and complex fluids[10], or solidification processing of metallic alloys[11–15]. The formation and evolution of boundaries in pattern-forming systems result from the different properties and mobilities of the patterns constituting the boundary. New regions may appear during the migration of a boundary, as in the case of freckles in metal casting[16] or that of invasion of cancerous tumors[6,7]. The formation and morphological evolution of these boundaries are in turn crucial for the behavior of biological systems[3–5] and the properties of structural materials[11–17].

In solidification processing, such as casting, welding, or additive manufacturing, cellular or dendritic patterns develop during the growth of a solid crystal from the liquid phase[10,13,15,18]. In metallurgy, each pattern region of similar crystallographic orientation is referred to as a grain. Each grain typically grows from a single solid seed and presents a distinctive growth direction. Grain boundaries (GBs) thus emerge during the growth competition between different grains with different pattern migration dynamics. The grain texture—the resulting macroscopic distribution of grain orientations and GBs—has a key influence on the material strength[17].

The relative orientation of grains is known to be a key determinant of the grain texture. The classical theory of GB evolution during polycrystalline solidification assumes that grains that have a crystalline orientation best aligned with the temperature gradient direction will progressively eliminate less aligned neighboring grains during growth[19,20]. Unexpected phenomena, however, such as overgrowth of favorably-oriented grains by unfavorably-oriented grains have been reported experimentally[21,22] and more recently studied theoretically and computationally. While those simulations have revealed a much more complex picture of grain competition than initially anticipated, they have mostly been confined to 2D or quasi-2D geometries[23–25]. In

[1]Department of Physics and Center for Interdisciplinary Research on Complex Systems, Northeastern University, Boston, MA, USA. [2]Materials Science Division, Lawrence Livermore National Laboratory, Livermore, CA 94550, USA. [3]Aix-Marseille Univ., Université de Toulon, CNRS, IM2NP, Marseille, France. [4]IMDEA Materials Institute, Getafe, Madrid, Spain. [5]Department of Material Science and Engineering, Iowa State University, Ames, IA, USA. ✉e-mail: a.karma@northeastern.edu

spatially extended 3D samples, the geometry of practical relevance, this competition has remained largely unexplored, for two main reasons. First, gravity-induced buoyancy in the liquid phase[26–28] has made it traditionally difficult to solidify samples under well-controlled homogeneous conditions, a requirement to elucidate grain competition mechanisms inherent to the non-equilibrium growth process. Second, simulations of alloy solidification in 3D spatially extended samples have remained computationally costly[29].

Herein, we address these challenges by combining experiments[18,30–33] performed in reduced gravity conditions onboard the International Space Station (ISS) with massively-parallelized quantitative phase-field (PF) simulations[25,34–38]. Our results shed light on the complex behavior of 3D GBs and their stability during polycrystalline solidification.

## Results and discussion

Experiments were performed in the Directional Solidification Insert (DSI) of the DEvice for the study of Critical LIquids and Crystallization (DECLIC) onboard the ISS. This setup allows imaging in situ the evolution of a solid-liquid interface during directional solidification of transparent organic compounds. The reduced gravity permits well-controlled and homogeneous conditions, with predominantly diffusive transport of heat and solute species, even in bulk 3D samples[30–34].

Figure 1 shows optical images of the solid-liquid interface during the growth of a transparent succinonitrile-0.24wt% camphor compound at an imposed growth velocity $V = 1.5\,\mu m/s$ within a temperature gradient $G = 19\,K/cm$, seen from a camera placed on the top of the

liquid and directly facing the advance of the solidification front. Shortly after the start of the experiment, the undercooled planar solid-liquid interface destabilizes and GBs appear, which are most noticeable from the distinct drifting directions of the different grains (Supplementary Movies 1–3). These GBs, almost straight initially (Fig. 1a), progressively develop finger-like protrusions when groups of cells penetrate the cellular pattern of the neighbor grain (Fig. 1b, c). This first in situ observation of such morphological roughening of the GB provides a visual explanation for grain morphologies previously observed in post-mortem characterization of fully solidified dendritic metallic microstructures[39].

A striking observation in these experiments is that the morphological instability of GBs may lead to grain interpenetration and to the invasion of a neighboring grain by a solitary cell (SC) after it detaches from its original grain (outlined in yellow in Figs. 1, 2). Several events of grain interpenetration and SC invasion were observed at a growth velocity $V = 1.5\,\mu m/s$ (Figs. 1, 2 and Supplementary Movies 1–3) but also $V = 2\,\mu m/s$ (Supplementary Figs. 1, 2 and Supplementary Movie 4). SCs typically emerge when a penetrating group of cells becomes disconnected from its original grain through the elimination of cells at the base of a finger-like protrusion by cells of the penetrated grain. Once detached from their original grain, SCs drift within the nearby (host) grain for several hours of experiments (Fig. 1a–c), but they can occasionally be eliminated, for instance when squeezed between two cells of different grains (Fig. 1d–h).

Figure 2 illustrates the time evolution of a SC within its host cellular pattern, namely within the orange box region of Fig. 1b, c. The SC

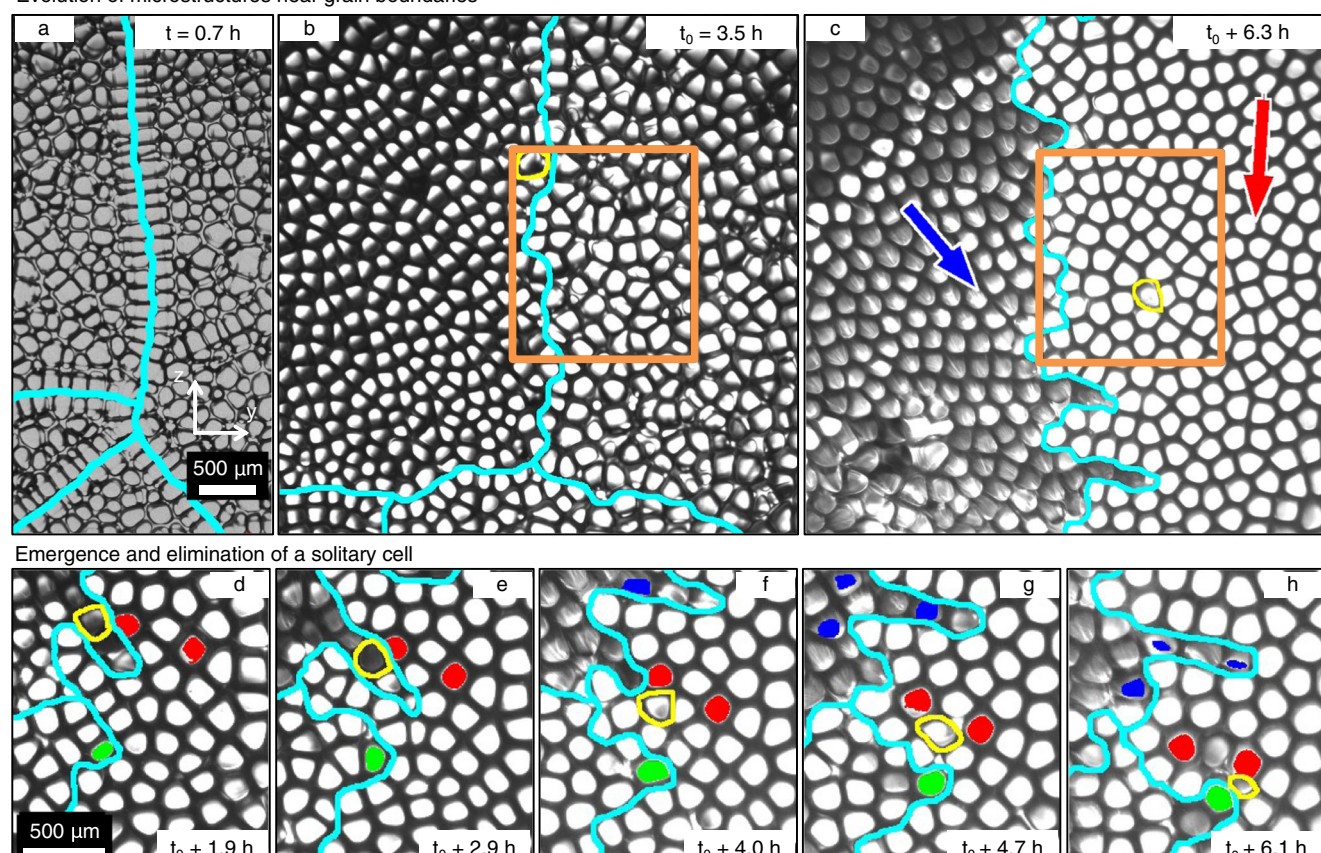

**Fig. 1 | Evolution of the solid-liquid interface pattern, grain boundaries, and emergence of solitary cells.** The interface is seen from its liquid side during directional solidification experiments in microgravity at different times *t*. Grain boundaries (GBs) appear in cyan lines. The GBs have a uniform linear morphology at early time in (**a**). A solitary cell (SC), outlined in yellow, emerges in the orange square of **b**, **c** (also shown in Fig. 2). Red and blue arrows in **c** indicate the overall drift directions of the grains. Another SC (outlined in yellow) in panels (**d**, **e**) emerges at $t_0 + 4.0$ h (**f**) and survives for a short time (**g**) before being eliminated (**h**). See Supplementary Movie 1 of (**a**–**c**), and 2 of (**d**–**h**).

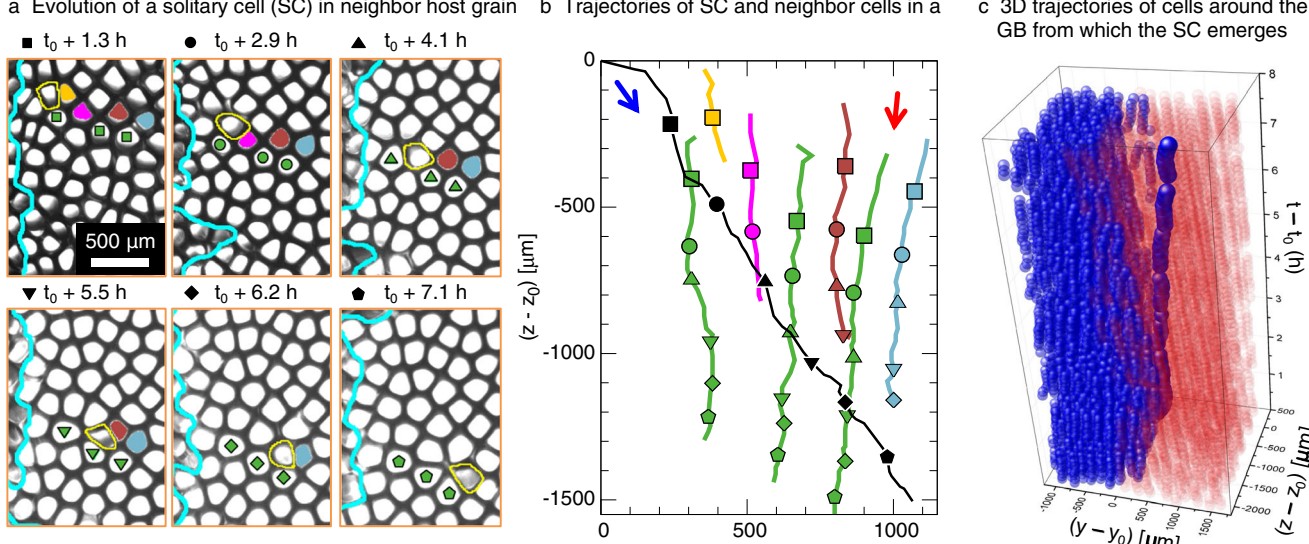

**Fig. 2 | Emergence and evolution of a solitary cell (SC) in DECLIC-DSI experiments.** Snapshots in (**a**) illustrate the evolution of a SC (outlined in yellow) and grain boundary (GB) morphologies (cyan lines). Trajectories of selected cells (black for the SC; green and other colors for tagged non-eliminated and eliminated host cells, respectively) appear in (**b**) with different symbols as time markers. Blue and red arrows in (**b**) indicate the overall drift directions of grains. Panel (**c**) shows cell trajectories in 3D, evidencing the GB roughening and branching and the resulting tubular defect inside the red grain. See Supplementary Movies 3 of (**a**), and 5 of (**c**).

emerges at a time $t = t_0$ (Fig. 1b) and drifts within its host grain until the end of the experiment at $t = t_0 + 7.6$ h (Fig. 2a). The observed trajectories of the SC and selected cells within the host grain appear in Fig. 2b, respectively in black line and colored lines corresponding to tagged cells in Fig. 2a.

Cells within the host grain move in an overall consistent direction (red arrow), but the motion of the SC remains prescribed by its own crystalline orientation (blue arrow). The evolution of the SC involves the elimination of cells within the host pattern (non-green trajectories in Fig. 2b) as well as accommodating the deformation of cells as they pass by each other without elimination (green trajectories). The drifting of a SC leads to the formation of an unexpected 3D branched structure consisting of a cylindrically shaped branch of approximately one cell diameter embedded in the host grain, as illustrated by the 3D reconstruction in Fig. 2c (Supplementary Movie 5). Such a tubular defect in the grain structure could have a major impact on the mechanical behavior of the microstructure.

We used a quantitative 3D PF model for dilute alloy directional solidification[25,35,40] to explore the mechanism of 3D branching of GBs (Fig. 1c) linked to the SC emergence. We simulated bi-crystalline microstructures that grow under the experimental conditions, i.e. considering a SCN-0.24wt% camphor alloy growing at $V = 1.5\,\mu m/s$ and $G = 19$ K/cm. Detailed simulation parameters are provided in Supplementary Methods. The only difference between the two grains is their crystal orientations. Each grain has a ⟨100⟩ preferred growth direction aligned along the axis $x'''$ (see below). This ⟨100⟩ direction makes an angle $\theta$ with respect to the $x$-axis (parallel to the $G$ direction), which can be inferred from the drift velocity of cells within a grain (see details in Supplementary Methods). The drift direction of those cells, as observed in the experimental movies, is defined by the projection angle $\phi$ of the $x'''$ axis onto the $y$-$z$ observation plane. We use $\phi_1$ and $\phi_2$, where the subscript stands for the two different grains (see below). Angles $\phi_1$ for grain 1 (blue) and $\phi_2$ for grain 2 (red) are measured from the $y+$ and $y-$ directions, respectively. Those drift angles were identified as key parameters for the occurrence of the invasion events observed in the experiments. The detailed relation between $(\theta, \phi)$ angles discussed here and the equivalent Euler angles $(\alpha, \gamma)$ used as input parameters to the phase-field simulations are fully detailed in the attached Supplementary Methods. As discussed below, these

simulations allowed identifying the critical importance of the crystalline orientations and of the primary array spacing (see Supplementary Notes 2–3) upon the morphological stability of the resulting GB.

First, we performed a spatially extended bi-crystalline simulation with similar conditions and crystal orientations as in the experiment of Figs. 1, 2, i.e. with $(\theta_1, \phi_1) = (6°, -56°)$ and $(\theta_2, \phi_2) = (3°, 84°)$ (see details on the identification of crystalline orientations from Supplementary Movies 1–3). Figure 3 shows the results of a simulation that started with a straight GB normal to the $y$-direction. Like in the experiment, the GB (cyan line in Fig. 3a) is morphologically unstable, as cells in the left (blue) grain tend to penetrate into the right (red) grain. The leader cell of an invading group, outlined in yellow in Fig. 3a, detaches and becomes a SC at $t = t_0$. Then, it progresses within the host grain for over 7.6 h (Supplementary Movie 6), like in the experiment. The trajectories of cells in the host grain (green lines and symbols in Fig. 3b) are consistent with the grain crystalline orientation (red arrow). Meanwhile, the SC trajectory (black line and symbols) follows its natural drifting direction dictated by the crystal orientation of its parent grain (blue arrow). The 3D mechanism of GB branching linked to SC emergence is illustrated in Fig. 3c, with a 2D slice through a plane containing the SC trajectory in Fig. 3d (Supplementary Movie 7). The resulting 3D roughening and branching of the GB are caused by the morphological instability of the GB and the subsequent SC emergence.

Experiments (Fig. 2) and PF simulations (Fig. 3) are in quantitative agreement with regards to the drifting direction and velocity of the SC. Yet, one interesting difference is that, in the experiments, the SC eliminates cells from the invaded grain along its path, while the SC in the PF simulation does not produce any cell elimination. We attribute this difference to the lower average cell spacing in experiments compared to PF simulations, in part due to the simplified frozen temperature profile assumption[36]. The lower cell spacing in experiments renders cells inside the invaded grain more susceptible to elimination as the local spacing fluctuates to accommodate the passage of the SC. This interpretation is supported by additional PF simulations that relate the SC fate to the local array spacing (see Supplementary Figs. 8, 9 and Supplementary Note 3).

Experiments only produce few large grains with different misorientations, and hence a small number of different GB types. Thus, we used PF simulations to explore more systematically the role of GB

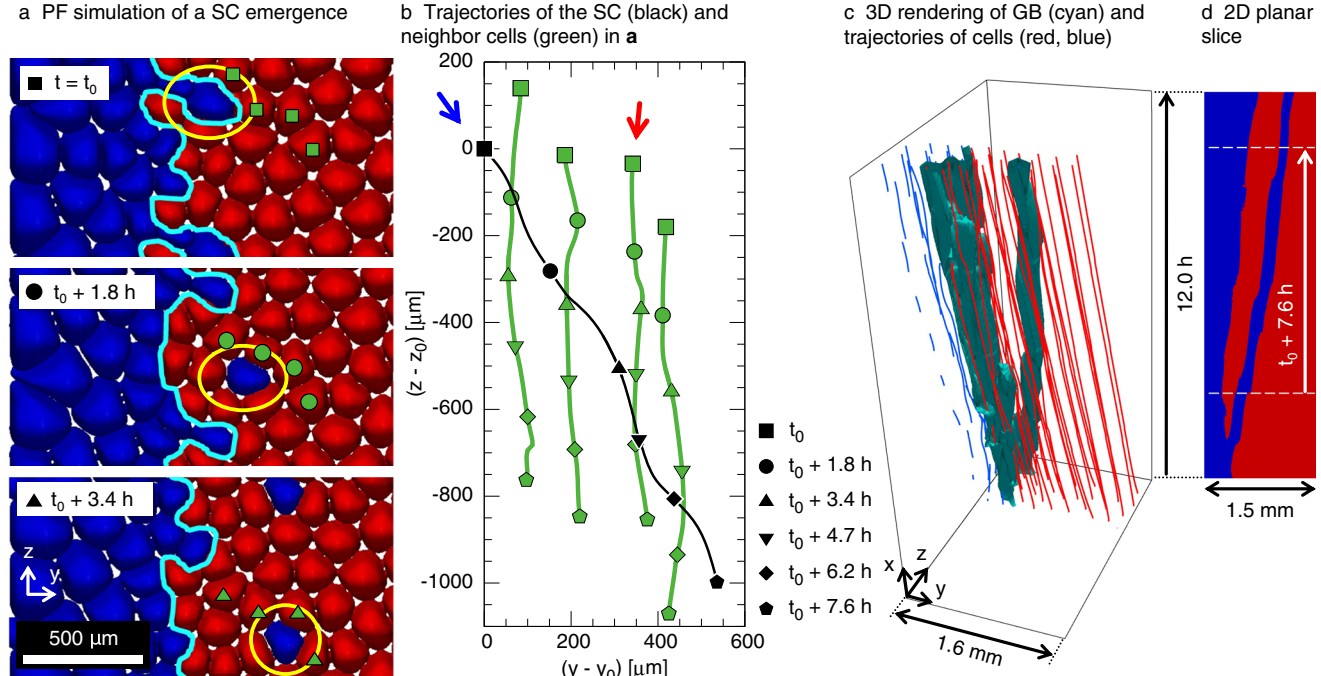

a  PF simulation of a SC emergence

b  Trajectories of the SC (black) and neighbor cells (green) in **a**

c  3D rendering of GB (cyan) and trajectories of cells (red, blue)

d  2D planar slice

**Fig. 3 | Emergence and evolution of a solitary cell (SC) in 3D PF simulation of DECLIC-DSI experiments.** Panel (**a**) (Supplementary Movie 6) shows the simulated pattern evolution for experimental parameters and grain orientations. For clarity, part of the images is duplicated vertically using the periodic boundary conditions. The emerging SC is outlined in yellow. Trajectories of selected cells (green) and the SC (black) over 7.6 h appear in (**b**), with symbols as time markers. Color arrows indicate drift directions of the blue and red grains. The 3D reconstruction of cell trajectories and the grain boundary (GB) in (**c**) (Supplementary Movie 7) and the 2D planar section in (**d**) show the progressive roughening and branching of the GB. Grain orientations are $(\theta_1, \phi_1) = (6°, -56°)$ and $(\theta_2, \phi_2) = (3°, 84°)$ as extracted from the analysis of the experimental videos (see details in Supplementary Methods).

bi-crystallography on the morphological stability of GBs. The results shown in Fig. 4 demonstrate that grain interpenetration and SC invasion occur over a wide range of misorientation angles that characterize the GB bi-crystallography.

Simulations were started with a straight GB normal to the *y*-direction, keeping a similar crystal tilt angle with respect to the temperature gradient $\theta = 5°$ for both grains (see definition of angles in Fig. 4a) and varying the azimuthal angles $\phi_1$ and $\phi_2$, corresponding to the pattern drift directions. We selected a low $\theta$ value because (i) only low tilt angles are observed experimentally, e.g. $\theta \approx 6°$ and $3°$ for the left and right grains in Fig. 1, respectively (see the method section and Supplementary Methods), and (ii) we found that GBs are usually morphologically stable for a large $\theta$. In particular, for a convergent GB between a well-oriented grain with $\theta_1 = \phi_1 = 0°$ and a misoriented grain of varying misorientation $\theta_2$ with $\phi_2 = 0°$, simulations reveal that the GB is only morphologically unstable for $\theta_2$ lower than about 15° (see Supplementary Fig. 6 and Supplementary Note 2). Moreover, while we primarily focused on the influence of the crystal orientations $(\phi_1, \phi_2)$ on the GB morphology, we also identified a key influence of the array spacing of initial microstructure upon the GB roughening dynamics. Indeed, cell invasion is promoted by a higher primary spacing within the host grain, as illustrated and discussed within Supplementary Note 3 (see Supplementary Figs. 8, 9).

Results of the mapping are summarized in Fig. 4b, with representative microstructures after 3 h of growth illustrated in Fig. 4c–f (also see Supplementary Movie 8). The entire competing dynamics of Fig. 4c–f can be found in Supplementary Movie 8. Inter-penetration (Fig. 4d) or one-sided penetration (Fig. 4e) of grains, marked as triangles in Fig. 4b, occur over a wide range of orientations. However, the invasion by a grain with $|\phi| \geq 75°$ was not observed. Hence, when both $|\phi_1|$ and $|\phi_2|$ are high, which corresponds to parallel drifting directions where the two grains slide against each other, the GB remains stable

(Fig. 4c and green squares in Fig. 4b). When grain penetration occurs, within the host grain, the leader cell of a penetrating group can either eliminate cells on its path or accommodate to squeeze itself between cells of the host array. After the leader cell clears a pathway, other penetrating cells follow, which results in the morphological destabilization of the GB. Following destabilization and penetration, this pathway may be interrupted by the elimination of one of the trailing cells by those of the host grain, hence leading to the isolation of an individual cell or group of cells. Events of SC emergence (Fig. 4f and circles in Fig. 4b) are only observed for $\phi_1 + \phi_2 \leq 90°$ when at least one grain penetrates into the other grain, i.e. for both $|\phi_1|$ and $|\phi_2|$ lower than 75°. This region (yellow background in Fig. 4b) contains the experimental conditions of Fig. 1 (black circle).

Even though this fingering instability of GBs was visualized here in situ for the first time in a transparent organic alloy, we expect it to occur during solidification of a wide range of technological metallic alloys, thereby potentially impacting mechanical behavior and other properties through the creation of complex 3D GB structures. Post-mortem metallographic analysis of Ni-based superalloys has revealed interpenetrating grain structures in serial sections[39], thus providing an indirect evidence of this instability in a well-developed dendritic growth regime. However, solidifying grains imaged in 2D through serial sectioning are usually reported to remain connected regions of space, i.e. with a GB evolution typically occurring by the progressive invasion of a pattern into its neighbor. Rapid advances in 3D and 4D (time-resolved 3D) imaging of metallic systems, e.g. using X-ray tomography, should yield further insight into the mechanisms of roughening, fingering, and solitary cell or dendrite emergence in technological metallic alloys. Beyond the field of solidification, we expect the present results to have potential relevance for the interpenetration of drifting cellular patterns in other contexts including biological systems.

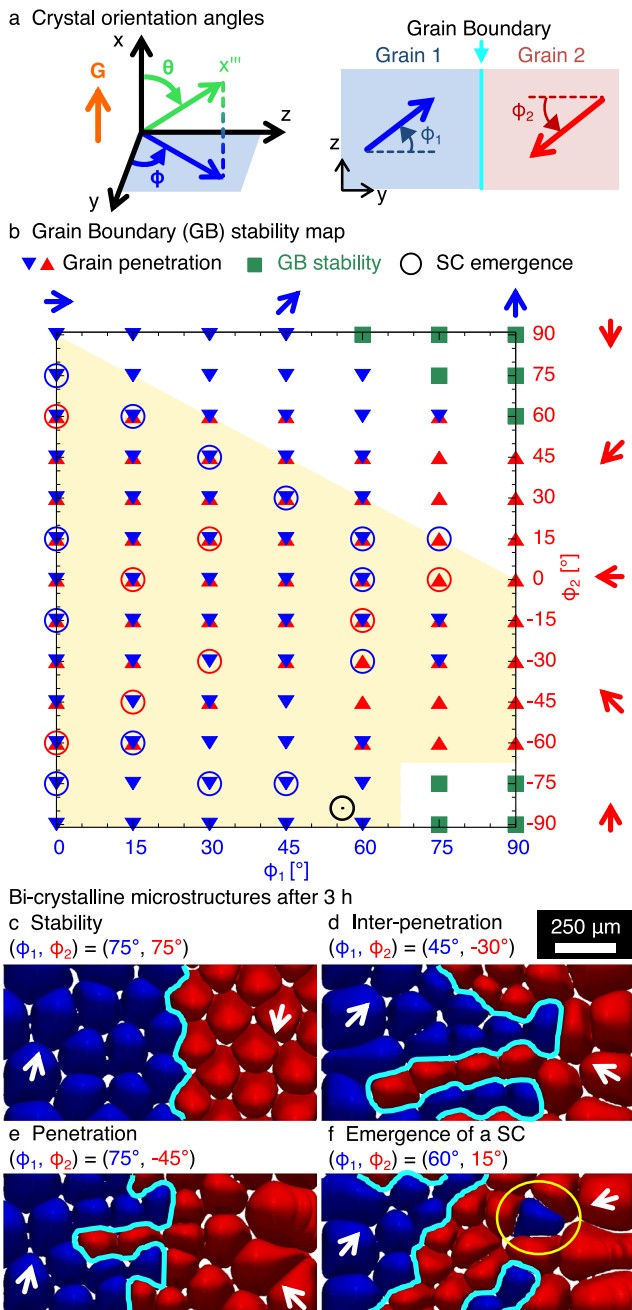

**Fig. 4 | Link between the stability of grain boundaries (GBs) and the crystal orientations.** Panel (**a**) shows the definition of crystal orientation angles $\theta$ and $\phi$. Subscripts 1 and 2 correspond to the left (blue) and the right (red) grains, respectively. Panel (**b**) (and Supplementary Fig. 7) shows the mapping of the convergent GB stability with $\theta_1 = \theta_2 = 5°$, which results in GB stability (green squares), grain penetration (triangles of the color of the penetrating grain), and/or solitary cell (SC) emergence (circles of the color of the emerging SC). The experimental SC is shown as the black circle. Arrows in (**b**, **c**) indicate drifting directions of grains. Panels (**c**–**f**) show representative microstructures after 3 h (Supplementary Movie 8).

temperature gradients $G$. The use of a transparent succinonitrile (SCN)-0.24wt% camphor alloy permits in situ optical imaging of the interface[31]. Experiments are typically initiated from a single crystal growing along the temperature gradient direction. Yet, after repeated experimental cycles of solidification and remelting of a given sample, polygonization of the seed may progressively develop, ultimately leading to the appearance of grain boundaries (GBs)[32,33].

The alloy solidifies within a cylindrical crucible with an inner diameter of 10 mm. The end temperatures of the setup are set higher and lower than the liquidus temperature of the alloy, such that the solid–liquid interface is located within a stable and well-controlled temperature gradient. The crucible is pulled through an axial adiabatic zone, allowing the interface to grow up to 100 mm in length at a controlled velocity. Thus, we can investigate the microstructure selection and image the in situ evolution of the solid–liquid interface while independently controlling the growth velocity $V$ and temperature gradient $G$ experienced by the interface.

Here we focus on the experiment with $G = 19$ K/cm and $V = 1.5\,\mu$m/s (and $V = 2.0\,\mu$m/s in Supplementary Figs. 1, 2), which yields a morphological destabilization of the grain boundary and the emergence of solitary cells. Further details on the experimental setup appear in previous publications[30–34].

## Phase-field simulations

We used a quantitative phase-field formulation for directional solidification[40,41]. The model assumes a one-dimensional frozen temperature profile, with a set temperature gradient $G$ moving at a constant velocity $V$ with respect to the sample, neglecting solute diffusion in the solid phase. The model was developed for exact asymptotic matching of the corresponding solid–liquid sharp-interface problem, even using a diffuse interface width much larger than its actual physical size[40,42]. The time evolution equation for the solute field includes an anti-trapping current across the solid–liquid interface in order to correct for the spurious numerical solute trapping arising from the use of a such a wide diffuse interface[40,43]. The model only uses one phase field for the two grains. The orientation of each grain is set by introducing an integer grain index field which prescribes the three-dimensional orientation of the grain[35].

Current simulations focus on one solidification condition, namely $V = 1.5\,\mu$m/s and $G = 19$ K/cm, for a SCN-0.24 wt% camphor alloy. Material and numerical parameters are similar to those used and discussed in previous publications[32,36] (Supplementary Table 2). The problem consists of two coupled partial differential equations, which are solved using finite differences on a homogeneous grid of cubic elements. We use a nonlinear preconditioning of the phase field[44], hence solving for the preconditioned phase field $\psi$ defined as $\varphi = \tanh(\psi/\sqrt{2})$, where $\varphi$ is the standard phase field that varies smoothly from $\varphi = +1$ in the solid to $\varphi = -1$ in the liquid. This simple nonlinear change of variable enhances the numerical stability of the simulations for larger grid spacings, and therefore allows faster (or larger) simulations with negligible loss in accuracy[30,34,44]. Detailed equations and relations between $\psi$, $\varphi$, and the two coupled partial differential equations are provided in Supplementary Methods as well as in previous works[30,34,36,44]. For numerical parameters, we use a diffuse interface thickness $W = 2.7\,\mu$m, a homogeneous grid of cubic elements of size $\Delta x = 3.2\,\mu$m, and an explicit time step of $\Delta t = 5.7$ ms. Thermal fluctuations are mimicked by the introduction of a random noise perturbation onto the preconditioned phase field $\psi$, with a dimensionless amplitude $F_\psi = 0.01$[34]. Simulations were parallelized on Graphics Processing Units (GPUs) using Nvidia™ CUDA programming platform.

The simulation domain size was $L_x \times L_y \times L_z\ [\mu m^3] = 2295 \times 1912 \times 633$ for Fig. 3a and $2295 \times 1272 \times 633$ for Fig. 4b, where $L_x, L_y$, and $L_z$ respectively correspond to spatial directions $x, y$, and $z$. We use periodic conditions along the $z$-normal boundaries, and no-flux

## Methods

### Microgravity experiments

Experiments were performed within the Directional Solidification Insert (DSI) of the Device for the study of Critical LIquids and Crystallization (DECLIC) aboard the International Space Station. Reduced gravity allows the processing of homogeneous 3D cylindrical samples using a classical Bridgman setup with various pulling velocities $V$ and

symmetric conditions for all other boundaries. The GB between the two grains is initially planar and normal to the $y$-direction. The GB of the simulation in Fig. 3a is initially located at a distance $L_y/4$ from the left ($y = 0$) boundary. This simulation over $t_{max} = 12$ h was performed in about 12 days ($\approx 290$ h) using eight Nvidia Tesla K80 GPUs. In Fig. 4b, the GB is initially located at the center of the domain in the $y$-direction. These simulations were performed for a total solidification time $t_{max} = 3$ h. Each one of the 69 simulations took between 150 and 170 h to perform on one Nvidia GeForce GTX Titan X GPU.

## Crystal orientation and drift velocity

When a crystal is misoriented with respect to the temperature gradient by an angle $\theta$, the resulting cellular or dendritic array drifts laterally with a lateral velocity $V_d$, projected on the experimental observation plane. Previous studies using a thin-sample geometry[45,46] have shown that the drift angle $\theta_d = \tan^{-1}(V_d/V)$ is typically lower than the crystal orientation $\theta$. The ratio between $\theta_d$ and $\theta$ is a function of the Péclet number $\mathrm{Pe} = \lambda V/D$, where $D$ is the liquid solute diffusivity (here $D = 270$ $\mu m^2/s$) and $\lambda$ is the array primary spacing, as:

$$\frac{\theta_d}{\theta} = 1 - \frac{1}{1 + f\,\mathrm{Pe}^g}. \qquad (1)$$

Parameters $f$ and $g$ depend upon the material. It was initially suggested[45,46] that $g$ was a constant while $f$ varied as a function of the grain angle $\theta$. However, we found good agreement with PF predictions throughout a wide range of orientations keeping both $f$ and $g$ constants[35].

We ascertained the validity of Eq. (1) in the current simulations and identified parameters $f = 0.67$ and $g = 1.47$ (see Supplementary Fig. 5 and Supplementary Note 1). To do so, we simulated hexagonal patterns (Supplementary Fig. 4a) with different primary spacings from $\lambda = 141$ to $320$ $\mu m$, with a screening step of about $22$ $\mu m$. On these structures, we imposed crystal orientations of $\theta = 5, 10,$ and $15°$ at $\phi = 0°$ (see Supplementary Fig. 5a). We also considered $\phi = 30°$ at $\theta = 10°$. To calculate $f$ and $g$ from these simulations, we measured the time evolution of the positions of the centers of cells in the $y$–$z$ plane every 333 s using a similar image processing method as described in a previous article[34]. We measured the drift velocity $V_d$ and thus the angle $\theta_d$, and then fitted simulation results to Eq. (1) for $f$ and $g$ (Supplementary Fig. 5d). We verified that this relation linking grain orientation and drift velocity was not only valid for hexagonal patterns (Supplementary Figs. 4a, 5a, and d), but also held for fcc-like arrays and pseudo-2D thin-sample confined arrays (Supplementary Figs. 4b, c, 5b, c, and e), as well as for different azimuthal orientation angles $\phi$ (Supplementary Fig. 5f), using the same values for $f$ and $g$.

Using Eq. (1) with these identified values of $f$ and $g$, we calculated the 3D crystal orientation of different grains in the experiments from the observed drifting velocities and directions. At $V = 1.5$ $\mu m/s$, the average primary spacing of the overall cellular array is $\lambda \approx 260$ $\mu m$. The drift velocities are $V_d = 0.09$ and $0.04$ $\mu m/s$ for the left-side and right-side grains in Fig. 1c, which yields $\theta_d = 3.4°$ and $1.5°$, respectively. Thus using Eq. (1) with the estimated constants, we calculate crystal angles $\theta = 6°$ for the left-side grain and $3°$ for the right-side grain. We then used these values for the imposed crystal angles in the PF simulation of Fig. 3a.

## Data availability

The authors declare that the data supporting the findings of this study are available within the paper and its supplementary information files.

## Code availability

The code for phase-field simulations is available from the corresponding author upon request.

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

## Acknowledgements

This study was supported by the National Aeronautics and Space Administration Grants NNX16AB54G, 80NSSC19K0135 and 80NSSC22K0664 (Y.S., K.J., R.T., A.K.), and the French Space Agency CNES project MISOL3D Microstructures de solidification 3D (F.M., B.B., N.B.). D.T. acknowledges support from the Spanish Ministry of Science via a Ramon y Cajal fellowship (RYC2019-028233-I). Y.S. acknowledges that this work was partially prepared by LLNL under Contract DE-AC52-07NA27344. The authors also gratefully acknowledge the access to the supercomputing Discovery cluster provided by Northeastern University.

## Author contributions

Y.S. and A.K. designed the simulations. Y.S., D.T., and K.J. implemented the simulation code. Y.S. and K.J. performed PF simulations. Y.S., D.T., and A.K. analyzed simulation data. F.M., N.B., B.B., and R.T. designed and remotely performed the DECLIC-DSI experiments. F.M. and N.B. analyzed experimental data. Y.S., D.T., F.M., K.J., N.B., and A.K. wrote the manuscript. All authors discussed the results and edited the manuscript.

## Competing interests

The authors declare no competing interests.
