## [Peer Review File · Nature Communications]

Cell invasion during competitive growth of polycrystalline solidification patternsREVIEWER COMMENTS

Reviewer #1:

Authors reported an important experimental finding that individual cells from one grain can unexpectedly invade a nearby grain of different misorientation, either as a solitary cell or as rows of cells, and have performed the phase field simulation to reproduce this phenomenon further demonstrating that invasion occurs for a wide range of misorientations. These results can compare with that post-mortem metallographic analysis of Ni-based superalloys has revealed interpenetrating grain structures. I think these results are of important significance and it is worth publishing. However, authors need classify the occurrence mechanism behind this phenomenon, and, technically, the following points should also be reflected in the paper.

1. Which parameter in the phase simulation affects the generation of the 'invasion' phenomenon? What cause is behind this?
2. The expressions of two basic anisotropic parameters alpha and gamma should be written in the manuscript.
3. What is the difference between the preconditioned phase field variable Psi to the phase field equation and concentration field equation and the phase field variable phi?
4. Page 17 in Supplementary Information, "We attribute this discrepancy between experiments and simulations to the underestimation of λ_{\min} in the simulations, due to the frozen temperature approximation". I think that the cause may be from the mathematical model. Authors should check their correctness of the model.
5. In figures, when (ϕ_1, θ_1) and (ϕ_2, θ_2) are changed, the morphological destabilization of the grain boundary appears. Do the boundaries of these grains share the same pair parameters?

Reviewer #2 (Remarks to the Author):

This article describes the results of directional solidification experiments in a dilute transparent succinonitrile-camphor alloy, conducted on the International Space Station. The experimental observations are supported by phase-field calculations that are able to reproduce some of the unexpected observations. The work should be of general interest to the scientific community, and clearly demonstrates the importance of combined experimental and theoretical/computational approaches.

The experiments in this article are a continuation of previous studies of cellular growth in this same apparatus and alloy system. Refs [28-30] describe oscillatory behavior in the cellular growth patterns when solidifying at 1 micron/s in a fixed gradient of 19 K/cm. The experiments in this article were conducted at a slightly higher speed, 1.5 micron/s, thus somewhat further from the marginal stability limit, where the initial grain pattern is further destabilized.

The article focuses on the competition of the growing cells near pre-existing grain boundaries. Several different phenomena are observed, ranging from maintaining of the prior boundary, invasion of one grain by another, and a rather surprising splitting off of one cell that continues as a “loner” into the neighboring grain. The authors use state-of-the-art phase field modeling to replicate all of these phenomena, demonstrating that the source of the various behaviors derives from the relative crystallographic orientation of the parent grains.

I have only one suggestion for a minor revision. I found Figure 4a difficult to understand. On the other hand, Figure SI4, in the supplemental material, contains essentially the same information, but in a much more easily understood form. I recommend replacing Figure 4a with Figure SI4, and perhaps adding a little prose to better describe the geometry and crystallography.

Responses to Reviewers

We thank both reviewers for their comments that have helped us to improve the quality of our manuscript. Our responses are given below. Green text corresponds to the reviewers' comments, black text corresponds to our replies, and blue text corresponds to added or modified text within the manuscript or SI document. Modifications are also highlighted as blue text within the revised manuscript and SI.

[Reviewer #1]

Authors reported an important experimental finding that individual cells from one grain can unexpectedly invade a nearby grain of different misorientation, either as a solitary cell or as rows of cells, and have performed the phase field simulation to reproduce this phenomenon further demonstrating that invasion occurs for a wide range of misorientations. These results can compare with that post-mortem metallographic analysis of Ni-based superalloys has revealed interpenetrating grain structures. I think these results are of important significance and it is worth publishing. However, authors need classify the occurrence mechanism behind this phenomenon, and, technically, the following points should also be reflected in the paper.

1. Which parameter in the phase simulation affects the generation of the 'invasion' phenomenon? What cause is behind this?

The simulation parameters intrinsic to the model (e.g. diffuse interface width) and the numerics (e.g. grid spacing) were selected after a thorough convergence analysis to ascertain that they do not affect the results. Therefore, within the discussion of the parameters influencing the invasion phenomenon, we can focus on physical parameters. In this study, we simulated one given alloy (SCN-0.24wt% camphor) and one set of processing parameters ($V = 1.5 \mu\text{m/s}$ and $G = 19 \text{ K/cm}$), such that we cannot unambiguously conclude on the effect of alloy or processing parameters on the invasion mechanism. However, we clearly identified two main aspects that critically affect the occurrence of the invasion mechanism, namely (1) the grains' orientations and (2) the array primary spacing.

In order to clarify this point, we added the following text:

As discussed below, these simulations allowed identifying the critical importance of the crystalline orientations and of the primary array spacing (see SI) upon the morphological stability of the resulting GB.

as well as:

Moreover, while we primarily focused on the influence of the crystal orientations (φ_1, φ_2) on the GB morphology, we also identified a key influence of the array spacing of initial microstructure upon the GB roughening dynamics. Indeed, cell invasion is promoted by a higher primary spacing within the host grain, as illustrated and discussed within SI (see Figs. SI8 and SI9 therein).

Moreover, we further clarified the invasion process by adding the following description in the text:

When grain penetration occurs, within the host grain, the leader cell of a penetrating group can either eliminate cells on its path or accommodate to squeeze itself between cells of the host array. After the leader cell clears a pathway, other penetrating cells follow, which results in the morphological destabilization of the GB. Following destabilization and penetration, this pathway may be interrupted by the elimination of one of the trailing cells by those of the host grain, hence leading to the isolation of an individual cell or group of cells.

2. The expressions of two basic anisotropic parameters alpha and gamma should be written in the manuscript.

The orientation of the grains can be expressed using classical Euler angles (α, γ) , as detailed in SI, or equivalently by angles (θ, φ) used in the main text. Although those two sets of angles are equivalent (they can be readily converted as explained in SI), we chose to refer only to the angles (θ, φ) in the main text for two main reasons. First, because they can be directly compared with experimental observations (φ appearing directly as the drift direction of cells within a grain), thus facilitating the interpretation and comparison between simulations and experiments. Second, we believe that presenting both sets of angles within the main text could result confusing for the reader, while not being necessary to the discussion. Hence, in the article, we use only (θ, φ) , illustrated in Fig. 4a, within the discussion.

We clarify the relation between (θ, φ) and (α, γ) in the revised manuscript as:

The only difference between the two grains is their crystal orientations. Each grain has a $\langle 100 \rangle$ preferred growth direction aligned along the axis x''' shown in Fig. 4a. This $\langle 100 \rangle$ direction makes an angle θ with respect to the x axis (parallel to the G direction), which can be inferred from the drift velocity of cells within a grain (see details in SI). The drift direction of those cells, as observed in the experimental videos, is defined by the projection angle φ of the x''' axis onto the y - z observation plane. As illustrated in the right panel of Fig. 4a, we use φ_1 and φ_2 , where the subscript stands for the two different grains. Angle φ_1 for grain 1 (blue) and φ_2 for grain 2 (red) are measured from the y^+ and y^- directions, respectively. Those drift angles were identified as key parameters for the occurrence of the invasion events observed in the experiments. The detailed relation between (θ, φ) angles discussed here and the equivalent Euler angles (α, γ) used as input parameters to the phase-field simulations are fully detailed in the attached SI.

3. What is the difference between the preconditioned phase field variable Psi to the phase field equation and concentration field equation and the phase field variable phi?

The preconditioned phase field ψ and classical phase field φ are related as $\varphi = \tanh(\psi/\sqrt{2})$. This simple nonlinear change of a variable was initially suggested by Glasner as a way to numerically stabilize the numerical resolution for coarser spatial grids (hence providing an additional advantage for upscaling of simulations) [*J. Comput. Phys.* 174, 695-711 (2001)]. The method has been incorporated in various of our previous phase-field studies [*Phys. Rev. E* 92, 042401 (2015)], [*Acta Mater.* 82, 64-83 (2015)], and [*Acta Mater.* 150, 139-152 (2018)] and has also been used by other groups (e.g. [*Ramirez & Beckermann, Acta Mater.* 53, 1721 (2005)] or [*Chadwick &*

Voorhees Acta Mater. 211, 116862 (2021)). The detailed equations for this preconditioned formulation were described in depth in previous articles.

To clarify this point, we have updated the method section of the article as:

We use a non-linear preconditioning of the phase field, hence solving for the preconditioned phase field ψ defined as $\phi = \tanh(\psi/\sqrt{2})$, where ϕ is the standard phase field that varies smoothly from $\phi = +1$ in the solid to $\phi = -1$ in the liquid. This simple nonlinear change of variable enhances the numerical stability of the simulations for larger grid spacings, and therefore allows faster (or larger) simulations with negligible loss in accuracy. Detailed equations and relations between ψ , ϕ , and the two coupled partial differential equations are provided in the SI as well as in previous works.

4. Page 17 in Supplementary Information, “We attribute this discrepancy between experiments and simulations to the underestimation of λ_{\min} in the simulations, due to the frozen temperature approximation”. I think that the cause may be from the mathematical model. Authors should check their correctness of the model.

The correctness of the phase-field model has been extensively tested in several previous studies, including the original studies where the quantitative phase-field formulation used here was first introduced [*Karma, Phys. Rev. Lett.* 87, 115701 (2001); *Echebarria, et al., Phys. Rev. E*, 70, 061604 (2004)]. This formulation uses an anti-trapping current to model the physically relevant limit of slow solidification for the present experiments with local thermodynamic equilibrium at the interface. The validity of the model was originally established by comparison with solutions of the corresponding set of sharp-interface equations (Gibbs-Thomson and Stefan interface conditions together with solute diffusion in the liquid) and by detailed convergence studies showing that the results are essentially independent of the width of the spatially diffuse interface region, as long as the diffuse interface width remain lower than the typical radius of curvature of the interface pattern. The same convergence tests were performed for the present simulations. The fact that the validity of this model was firmly established in those original studies, and that similar convergence studies have been performed by other research groups, has contributed to its wide use in the solidification community (as the high number of citations to those studies testifies).

Moreover, we have used this model to reproduce a number of other observations from microgravity experiments performed within the same setup and the same SCN-camphor alloy as in the present study, such as oscillatory modes [*Bergeon, et al., Phys. Rev. Lett.* 110, 226102 (2013); *Tourret, et al., Phys. Rev. E* 92, 042401 (2015); *Pereda, et al., Phys. Rev. E*, 95, 012803 (2017)] and primary spacing selection [*Mota, et al., Acta Mater.* 204 116500 (2021)]. The fact that the same model was capable of capturing complex mechanisms with remarkable accuracy (e.g. predicting breathing modes oscillations of period 48.1 min compared to experimentally measured period of 45.6 min [*Phys. Rev. Lett.* 110, 226102 (2013)]) provides, in our view, sufficient confidence regarding the predictive capabilities of the model.

Rather than the quantitative validity of the phase-field model, we believe that the quantitative discrepancy between observed and simulated primary spacings originates from uncertainties in thermal conditions and also potentially material parameters. The temperature gradient is not directly measured during the microgravity experiments and needs to be inferred from a thermal calculation that takes into account the entire crucible geometry. The release of latent heat, not

included in the present calculations for computational efficiency, has been shown previously to lead to a modest increase of the dynamically selected primary spacing [Mota, et al., *Acta Mater.* 85, 362-377 (2015); Song, et al., *Acta Mater.* 150, 139-152 (2018)]. Finally, while all material parameters are measured, those measurements also contain uncertainties that may contribute to quantitative differences between modeling and experiment. In our view, the fact that the model reproduces complex phenomena such as grain interpenetration and cell invasion, despite the quantitative difference between observed and simulated spacings, demonstrates that those phenomena are robust and not too sensitive to material parameters or growth conditions within reasonable ranges.

We have revised the relevant text (SI, page 17) to better explain the possible contributing factors underlying the quantitative discrepancy between observed and measured spacing, as follows:

We attribute this discrepancy between experiments and simulations to potential uncertainties in material properties and thermal conditions, which are not directly measured during the experiments. For computational efficiency, present calculations consider a frozen temperature approximation (i.e. a one-dimensional temperature field with constant G and V) and do not include the effect of latent heat release during solidification. Yet, previous studies have shown that relaxing these approximations may lead to a modest increase of the dynamically selected primary spacing [Mota, et al., *Acta Mater.* 85, 362-377 (2015); Song, et al., *Acta Mater.* 150, 139-152 (2018)].

5. In figures, when (ϕ_1, θ_1) and (ϕ_2, θ_2) are changed, the morphological destabilization of the grain boundary appears. Do the boundaries of these grains share the same pair parameters?

All simulations in the article are performed using the same parameters except for the set of crystal orientations (θ_1, ϕ_1) of grain 1 (in blue) and (θ_2, ϕ_2) for grain 2 (in red). Simulations are set up to either use the set of angles assessed for experimental observations (Fig. 3), or a broad range of (ϕ_1, ϕ_2) while fixing $\theta_1 = \theta_2 = 5^\circ$ (Figs 4, SI7, and SI8), or exploring different values of θ_2 for fixed $\theta_1 = \phi_1 = \phi_2 = 0^\circ$ (Fig. SI6). We clarified it in the manuscript and SI by adding the grains' orientations within the caption of every figure where it was not already explicitly mentioned.

[Reviewer #2]

This article describes the results of directional solidification experiments in a dilute transparent succinonitrile-camphor alloy, conducted on the International Space Station. The experimental observations are supported by phase-field calculations that are able to reproduce some of the unexpected observations. The work should be of general interest to the scientific community, and clearly demonstrates the importance of combined experimental and theoretical/computational approaches.

The experiments in this article are a continuation of previous studies of cellular growth in this same apparatus and alloy system. Refs [28-30] describe oscillatory behavior in the cellular growth patterns when solidifying at 1 micron/s in a fixed gradient of 19 K/cm. The experiments

in this article were conducted at a slightly higher speed, 1.5 micron/s, thus somewhat further from the marginal stability limit, where the initial grain pattern is further destabilized.

The article focuses on the competition of the growing cells near pre-existing grain boundaries. Several different phenomena are observed, ranging from maintaining of the prior boundary, invasion of one grain by another, and a rather surprising splitting off of one cell that continues as a “loner” into the neighboring grain. The authors use state-of-the-art phase field modeling to replicate all of these phenomena, demonstrating that the source of the various behaviors derives from the relative crystallographic orientation of the parent grains.

I have only one suggestion for a minor revision. I found Figure 4a difficult to understand. On the other hand, Figure SI4, in the supplemental material, contains essentially the same information, but in a much more easily understood form. I recommend replacing Figure 4a with Figure SI4, and perhaps adding a little prose to better describe the geometry and crystallography.

As suggested, we updated Fig. 4a replacing it by the supplementary Fig. SI4.

We also revised relevant text in order to better describe the geometry and crystallography:

We simulated bi-crystalline microstructures that grow under the experimental conditions, i.e. considering a SCN-0.24wt% camphor alloy growing at $V = 1.5 \mu\text{m/s}$ and $G = 19 \text{ K/cm}$. Detailed simulation parameters are provided in SI. The only difference between the two grains is their crystal orientations. Each grain has a $\langle 100 \rangle$ preferred growth direction aligned along the axis x''' shown in Fig. 4a. This $\langle 100 \rangle$ direction makes an angle θ with respect to the x axis (parallel to the G direction), which can be inferred from the drift velocity of cells within a grain (see details in SI). The drift direction of those cells, as observed in the experimental videos, is defined by the projection angle φ of the x''' axis onto the y - z observation plane. As illustrated in the right panel of Fig. 4a, we use φ_1 and φ_2 , where the subscript stands for the two different grains. Angles φ_1 for grain 1 (blue) and φ_2 for grain 2 (red) are measured from the y^+ and y^- directions, respectively. Those drift angles were identified as key parameters for the occurrence of the invasion events observed in the experiments.

and

, i.e. with $(\theta_1, \varphi_1) = (6^\circ, -56^\circ)$ and $(\theta_2, \varphi_2) = (3^\circ, 84^\circ)$ (see details on the identification of crystalline orientations from experimental videos in the SI).

as well as, in SI:

In single-grain simulations, φ is the (counterclockwise) angle with respect to the y^+ direction. In bi-crystalline simulations, the orientation φ_1 of grain 1 (represented in blue) is measured with respect to the y^+ direction (as in single-grain simulations), while the orientation φ_2 of grain 2 (represented in red) is measured from the y^- direction, both of them measured counterclockwise, as illustrated in Fig. 4 in the Letter.

We also added the grains' orientations within the caption of every figure where it was not already explicitly mentioned.

REVIEWERS' COMMENTS

Reviewer #1 (Remarks to the Author):

The reviewing comment for the article NCOMMS-22-48709A.

The authors investigated the Cell invasion during competitive growth of polycrystalline solidification, based on the experiment on the International Space Station. With the phase-field calculations, authors reproduced some of the unexpected observations and the nonlinear competition phenomenon, and demonstrate the source of the growing cells, either as a solitary cell or as rows of cells.

This work is interesting, systematic, and therefore it is recommended to publish.

Mingwen Chen

Reviewer #2 (Remarks to the Author):

The authors have responded in a clear and complete manner to the comments and questions of the reviewers. This work is interesting to the materials science community, and I highly recommend that it be accepted for publication in its present form.